# Detection of Human Gait Phases Using Textile Pressure Sensors: A Low Cost and Pervasive Approach

**DOI:** 10.3390/s22082825

**Published:** 2022-04-07

**Authors:** Matko Milovic, Gonzalo Farías, Sebastián Fingerhuth, Francisco Pizarro, Gabriel Hermosilla, Daniel Yunge

**Affiliations:** School of Electrical Engineering, Pontificia Universidad Católica de Valparaíso, Av. Brasil 2147, Valparaíso 2362804, Chile; matko.milovic@pucv.cl (M.M.); gonzalo.farias@pucv.cl (G.F.); sebastian.fingerhuth@pucv.cl (S.F.); francisco.pizarro@pucv.cl (F.P.); gabriel.hermosilla@pucv.cl (G.H.)

**Keywords:** textile sensors, gait analysis, supervised machine learning, smart clothes, multivariate time series classification, data annotation

## Abstract

Human gait analysis is a standard method used for detecting and diagnosing diseases associated with gait disorders. Wearable technologies, due to their low costs and high portability, are increasingly being used in gait and other medical analyses. This paper evaluates the use of low-cost homemade textile pressure sensors to recognize gait phases. Ten sensors were integrated into stretch pants, achieving an inexpensive and pervasive solution. Nevertheless, such a simple fabrication process leads to significant sensitivity variability among sensors, hindering their adoption in precision-demanding medical applications. To tackle this issue, we evaluated the textile sensors for the classification of gait phases over three machine learning algorithms for time-series signals, namely, random forest (RF), time series forest (TSF), and multi-representation sequence learner (Mr-SEQL). Training and testing signals were generated from participants wearing the sensing pants in a test run under laboratory conditions and from an inertial sensor attached to the same pants for comparison purposes. Moreover, a new annotation method to facilitate the creation of such datasets using an ordinary webcam and a pose detection model is presented, which uses predefined rules for label generation. The results show that textile sensors successfully detect the gait phases with an average precision of 91.2% and 90.5% for RF and TSF, respectively, only 0.8% and 2.3% lower than the same values obtained from the IMU. This situation changes for Mr-SEQL, which achieved a precision of 79% for the textile sensors and 36.8% for the IMU. The overall results show the feasibility of using textile pressure sensors for human gait recognition.

## 1. Introduction

Gait analysis is a process in which measurements are recorded and interpreted to characterize how a patient walks. Through this analysis, a health professional can make decisions on the patient in treatment [1,2]. For example, gait analysis tools can help distinguish between different diseases [3], evaluate an injury’s severity or extension, monitor the patient’s surgery recovery, or assess whether a kinesiology treatment has been effective for the patient or not [4]. Additionally, gait analysis is important to detect early neurological or musculoskeletal diseases [5]. Therefore, gait analysis has a wide field of application [6,7].

The human gait cycle is characterized by a step succession in time that involves many body movements. This cycle is analyzed separately by each foot or leg, dividing it into two main phases: the support phase, which corresponds to the time that the foot is in contact with the ground, and the swing phase, which consists of the time in which the foot is in the air. Both phases have several sub-phases, which are also usually integrated into the gait analysis, together with the measurement of other spatiotemporal parameters, such as the stride’s length or width, among others [8,9]. Different measurement systems can obtain spatiotemporal parameters [10]. Nowadays, the most widely used system for this is based on motion captures using arrays of infrared cameras [11,12] since they provide a three-dimensional analysis based on reliable and precise measurements of the human movement [5,7]. However, these systems have drawbacks, such as high implementation costs and the need for expert supervision, reaching fees greater than USD 30,000 [13] for the patients. Moreover, several authors have pointed out that camera-based analyses negatively impact the representativeness of the measurements compared to real-world situations because the patient is aware of being observed [13,14,15,16].

Recent research has focused on obtaining these spatiotemporal parameters of interest through wearable devices that, unlike infrared cameras, can be used both inside and outside the medical facility [17,18]. In addition, they usually have lower prices [13] because they are based on cheap MEMs inertial technology, such as accelerometers, gyroscopes, magnetometers, or the combination of those [19,20,21]. However, inertial measurement units (IMUs) and accelerometers have intrinsic drift errors that grow over time as a result of the continuous integration of the signals when the limb positions or angles are calculated [22]. Some solutions that include multiple body sensor(s) (BSN) powered by IMUs can accurately measure spatiotemporal parameters and leg flexion angles. However, the stiffness and size of the holding printed circuits boards (PCBs) used by the IMUs limit the pervasiveness of the solution, commonly limiting the free movement of the patient when walking. Moreover, even though recent commercial products have made improvements in that regard, such as the MEVA e-skin pants, from the manufacturer Xenoma [23], their acceptance among older adults, the main target population for these devices, is debatable [24].

The research on sensing wearables, such as insoles to measure foot pressure [25], or smart clothes [26,27], has increased in the last years with great results. This type of approach has the advantage of being more pervasive and appropriate for applications where it is necessary to measure continuously or outside the laboratory. Kim et al. [28] developed a full-body motion-sensing suit based on stretch sensors, which can be suitable for gait analysis applications. Nie et al. [29] developed a smart insole with an array of textile pressure sensors to map plantar stress distribution, which can be used for human gait analysis. Aqueveque et al. [30] studied the use of two templates with capacitive pressure sensors, successfully segmenting phases and sub-phases of the gait cycle in test participants. Furthermore, Lou et al. [31] manufactured a new graphene-based pressure sensor to measure plantar pressure in gait analysis. On the other hand, flexible pressure sensors have been used to attach to the legs to measure muscle activity and motion detection [32,33], but not yet for gait analyses.

This work aims to evaluate the use of low-cost and easy-to-build textile pressure sensors in human gait analysis applications, focusing on recognizing each leg’s swing and stance phases. This goal was achieved by assembling and fixing the flexible pressure sensors to auxiliary expandable fabric pants, distributing them equally on the muscles of the legs and knees. This pant provides a pervasive solution that can be used indoors and outdoors.

Aside from gait analysis, textile sensors have benefits over IMU sensors in other applications, such as preventing ulcers of bedridden patients by measuring the exact spot where excessive or permanent force is applied. IMUs are not appropriate for this application since they measure acceleration and not pressure. Other similar uses of textile sensors might include lifestyle applications to let users know if they have been seated too long, during office work, or driving.

Low-cost sensors are more suitable for continuous analysis during long periods (weeks, months) using pants since users require many changes of clothes for hygiene reasons. In addition, users are more prone to frequently use low-cost wearable devices without wear and tear concerns than expensive ones. Low-cost sensors also enable multiple sensing points in the pants, increasing the ability of the system to detect specific gait phases. In contrast, other solutions might not require multiple sensing points, e.g., sensors that are located under the soles of the feet, such as insoles, shoes, or socks. These sensors have great results in gait analysis due to the direct measure of the physical phenomenon associated with walking; however, they might have trouble detecting other gait phases or parameters, such as flexion angles. Accordingly, we aim at using multiple textile sensors to estimate flexion angles of each leg simultaneously without the need for IMUs. Nevertheless, this will be part of future work since this task requires more sophisticated machine learning techniques than those used in this work, given the nonlinear behavior of sensors, as explained below. At this moment, our proposed method is meant to complement the traditional solutions mentioned above, aiming to advance in fully wearable applications, to help patients with mobility impairments, e.g., elderly people, in a pervasive way.

Nevertheless, inexpensive textile sensors are prone to manufacturing tolerances, leading to sensitivity variations among sensors of the same characteristics. This drawback would exclude such sensing devices from precision-demanding applications, as those in the medical domain. However, commercial wearable applications, such as StretchSense products [34], demonstrate that textile sensors combined with machine learning (ML) methods can be helpful, even though this particular manufacturer focuses on high precision motion capture of hands. Therefore, their technology is more sophisticated and expensive than the proposed method.

ML methods can learn characteristics from various physical systems. Accordingly, we evaluated different ML algorithms for multivariate temporal signals—namely, random forest, time series forest, and Mr-SEQL—on the signals generated by our textile sensors to conduct a primitive gait analysis process. The analysis consisted of classifying three different gait cycle phases.

In addition, we evaluated the performance of the textile sensors and the ML algorithms by conducting a trial on three test subjects. The subjects also used IMU sensors on the tests, whose signals were processed with the same ML algorithms for comparison purposes. The experimental results showed that the estimations of the gait phases ere classified similarly for both types of sensors, so the first contribution of this work is the evaluation that shows that the use of low-cost textile technology is feasible for mobility monitoring of people in a pervasive fashion.

The second contribution of this work consists of introducing a new method for real-time labeling datasets generated from sensors related to body part movements, such as accelerometers, pressure sensors, IMUs, gyroscopes, or barometers. Datasets are necessary to train ML algorithms or models. In the case of supervised learning of temporal signals, datasets are created from the signals the ML models will learn from but also require a set of labels, which indicates the models in which characteristics correspond to the signals in a particular moment. In our case, these datasets are used to train the learning models to recognize a person’s movements. The problem is that creating the set of labels can be tedious since this task must be done manually.

For this reason, the proposed automatic labeling method uses a camera that captures the participant’s movements, whose images are fed into a pose estimation model called BlazePose [35], based on deep learning. Fortunately, the pose model is already trained and available and returns 3D coordinates of the different parts of the participant’s body. BlazePose allows programmers to easily understand and make decisions from the subject’s position by comparing the coordinates of particular body parts in the model, such as the knees or the feet. In our case, this feature allowed us to establish decision rules for the automatic labeling of the classes based on data, such as the velocity of the toes and heels. As a result of the labeling process, the corresponding class is printed and time-stamped in a text file whenever the coordinate set matches any rule. Later, this file is synchronized with the sensors’ measurements, which in our case are obtained from a wireless communicating acquisition board attached to the stretch pants at the waist.

Source code and schematics are available in a GitHub repository [36]. Datasets are available upon request to the corresponding author due to privacy restrictions.

## 2. Materials and Methods

This section describes the textile pressure sensors used in this work and the experimental setup to acquire the sensor signals. We also show how the study participants were selected and the protocol to obtain the data for the study. Finally, we present the workflow for the data treatment and the machine learning model evaluation.

### 2.1. Textile Pressure Sensor

In this study, we used resistive textile pressure sensors based on the procedure and design outlined in [37]. Ten sensors were manufactured using office supplies and from low-cost materials, such as (a) anti-static black bags (made of a low-density polyethylene (LDPE) sheet (ANT006BCB)), (b) conductive Shieldex NoraDell woven fabrics, (c) conductive threads Shieldex 117/17 DTEX, and (d) a plastic sheet, to cover the sensor with a thermo-laminate, as shown in Figure 1a. The manufacturing process for the sensors was quick, requiring less than ten minutes each. Then, we evaluated each sensor based on its response to a broad range of pressures due to the dependence of its electrical characteristics on the manufacturing process [38]. They had an approximate recovery time of 17 s for pressures below 10 kPa. They also presented reasonably linear resistance variations for pressures between 1 and 70 kPa, making them good candidates for motion detection or contact detection applications. In such applications, changing pressures are measured; therefore, knowing the trend of the measured signal is more relevant rather than a precise value [37].

### 2.2. Experimental Setup

For this study, we strategically distributed the textile pressure sensors over different points of both legs and attached them to expandable fabric pants, as shown in Figure 2. They were glued and sewn to it, resulting in the prototype shown in Figure 1b. The body parts chosen were vast hamstrings, quadriceps, femoral biceps, gastrocnemius, and patellas, because those are locations where the muscles can perform greater pressure on the sensors. While this amount is arbitrary and may be excessive, it is expected that they will generate enough spatial coverage in the pants to detect all types of leg movements. The purpose is for sensors to respond adequately to individual movements of each part of the body. Another reason is to associate the sensors’ pressures to the angles generated when the patient flexes his legs. The expandable fabrics pants used in this study were composed of 61% polyamide, 30% polyester, and 9% elastane, and are known as “base layer”, a kind of winter underwear.

These sensors were connected to an Arduino-based acquisition board, which was placed in the right buttocks area for more comfortable wear (see Figure 3b). Each sensor was sewn via two conductive threads through the pant fabrics to connect them with the circuit. This acquisition board, which is shown in Figure 3a, connects the textile part of the prototype and receives the measurements from the textile pressure sensor arrangement through a 74HC4067 16-channel multiplexer. At the same time, an MPU9250 IMU sensor was included in the electronics for comparison reasons with the signals from the textile sensors. As mentioned, using an IMU sensor is standard for the activity assessment of patients, as it captures physical variables associated with the patient’s movements and positions, such as the axial components of acceleration (expressed in m/s${}^{2}$), angular velocity (in rad/s), and the magnetic orientation (in μT). The microcontroller selected for the acquisition board was an STM32F103C8T6, allowing measurements with a sampling rate of 100 Hz. It communicates via UART with an HC-06 Bluetooth module, which wirelessly transmits the samples to a PC, storing them in a text file using a standard terminal program.

Additionally, a webcam recorded 1080p videos at 60 FPS, which served a dual purpose. On the one hand, it served to register the experiment, to associate the temporal signals from the sensors to the test participants’ movements. On the other hand, it enabled the automatic real-time labeling system, which is described later in Section 2.5.1.

### 2.3. Participants

Three volunteers participated in the trials to evaluate the ML algorithms. Their weight, height, age, and BMI are presented in Table 1. At the moment of the trials, these three volunteers had an average age of 27, an average weight of 75 kg, an average height of 1.70 m, and an average BMI of 25.7. The criterion for selecting the volunteers was that they had no mobility impairments.

### 2.4. Study Protocol

Each participant used the pants with a sensor distribution, as shown in Figure 2. At the beginning of the trial, the participants positioned themselves within the webcam’s frame. They walked around for 15 min in a straight 3.5 m path (see Figure 4d), moving without interruptions and at their own pace. Moreover, before starting the trial, the participants were asked to perform three short jumps, which served as synchronization points between the webcam recordings and the measurements acquired from the sensors.

### 2.5. Workflow

Generated datasets were processed as in the workflow shown in Figure 5. The main blocks are (a) labeling of the sensor data, (b) data pre-processing, (c) feature extraction and selection, and (d) the machine learning algorithms. In the end, the algorithms were benchmarked.

#### 2.5.1. Data Labeling

Since one research goal was to evaluate the ability of flexible pressure sensors to identify the phases of swing and support in each leg, each event was assigned one of the three possible classes based on the combinations of the phases that could be found during the walk:RSWLST: right foot swing and left foot stand (see Figure 4a);RSTLSW: right foot stand and left foot swing (see Figure 4c);RSTLST: stand on both feet (see Figure 4b).

A fourth “not considered” class called “NC” was assigned to those instances where measurements were recorded out of the 15% horizontal limits of the frame, as participants used to reduce their speed in those regions to turn and walk in the opposite direction. This dis-acceleration at the end of the runway did not correctly represent the body movements of a regular walk.

The event labeling process was carried out by executing a real-time pose estimation model, called BlazePose, on webcam recording of the walk, which located 33 different points of the human body [35].

Human gait analysis is based on gait phase recognition, identifying the moments when a transition occurs between stand and swing phases. Therefore, the heel and toe contact with the floor takes relevance. For each foot, the velocity at the heel and toe was estimated using a Python code, calculating the variation in position within the frame as a function of time of the current sample, compared to the previous one. These speeds will be adjusted based on an approximate walking pace of 0.7 m/s and a fixed distance between the camera and the runway.

The system allows generating labels of the three classes at a sample rate of 10 Hz, relating each of the labels with a timestamp measured in seconds of program execution. Both data are stored in a text file at the end of the code execution. Because this labeling system is not robust yet, the generated labels were reviewed by visual inspection of the videos, and misclassifications were fixed. The performance of this system is evaluated in Section 3.2.

It is worth mentioning that this labeling rate is ten times lower than the sampling rate of the sensing data at the acquisition board. Therefore, once labels were generated and the sensor measurements were acquired, both timestamps were synchronized using the jumps performed by the participants as a reference point. Labels were interpolated after this was done, associating each sensor measurement to a class.

#### 2.5.2. Data Pre-Processing

The next step in the workflow was performed in MATLAB. It consisted of pre-processing the multiple raw data channels generated by the participants’ sensors during the tests. This step was necessary to properly condition the signals before using them in the next characteristic extraction stage. The pre-processing stage included five parts, as described and listed below.

Every data instance labeled as “NC” was removed from the dataset since they did not represent a regular walk in the participants’ gait cycles;Any instance that had missing or incomplete data due to transmission failures during the acquisition was deleted;The outliers of each signal were identified using the isoutlier function of MATLAB. This value was replaced with the average plus the standard deviation;A fifth-order low-pass Butterworth filter with a cutoff frequency of 20 Hz was applied to filter out the high-frequency noise in each signal;The signals from each sensor were individually normalized using the MATLAB function normalize, which centers the data vector at $\bar{x}$ = 0 and a standard deviation σ = 1.

Once all signals went through these steps, the signals from the IMU and the textile pressure sensors were separated into two different files, generating two datasets with the same time stamps and labels and storing them in a file in ARFF format.

#### 2.5.3. Feature Extraction and Selection

For the selection of characteristics, we applied an automated extraction method using the Python package, called tsfresh [39], which integrates 63 different time signal characterization methods and allows to calculate a maximum of 794 characteristics of time series. This package is integrated into the Python library named Sktime [40], being used not only for features extraction but also to select the features that will be more representative for the type of data used by the function TSFreshRelevantFeatureExtractor. As a result, 270 features were obtained from the textile sensor set, including characteristics in the time and frequency domains. In contrast, for the datasets created with the IMU signals, 269 features were selected.

#### 2.5.4. Machine Learning Algorithms

In order to evaluate the performances of the sensors for the recognition of the swing and stance phases of each leg, three standard machine learning algorithms used in the multivariate time series were benchmarked [41].

Random forest: this algorithm is characterized by deciding which class an entry corresponds to when evaluating a set of randomly generated and trained decision trees [42];Time series forest (TSF): this classification algorithm is a meta-estimator and variant of the random forest algorithm for time series data. The data fit several decision tree classifiers on various sub-samples of a transformed dataset. It uses averages to improve predictive accuracy and control overfitting. For this algorithm, the sub-sample size is always the same as the original input sample size, but samples are drawn with replacements [43];Mr-SEQL: this algorithm is used to classify the univariate time series to train classification models (logistic regression) with characteristics extracted from multiple symbolic representations of time series (SAX), extracting features through the use of SEQL [44]. This method can be used for multivariate, such as ours, using a column assembly method.

#### 2.5.5. Performance Evaluation

The measurements obtained using the flexible pressure sensors and the IMU were synchronized temporally with the labels generated with the BlazePose method, creating eight datasets. These datasets were used later to train and test the classification models trained with different machine learning algorithms, mentioned in Section 2.5.4. Four of the eight datasets included only the flexible pressure sensor measurements and, for the other four, the IMU measurements. From one of these four datasets, three matched to the measurements of each test participant separately, and in the fourth, samples from the three pooled test participants were evaluated in a single dataset.

The performance evaluation of the classification models was carried out based on standard metrics used in machine learning, namely, the accuracy and confusion matrices. This way, it is possible to validate the results obtained with the flexible pressure sensors by contrasting them with those obtained with the IMU. In addition, it can be evaluated if the use of the flexible sensors requires an individual calibration process for each participant that uses the smart pants.

## 3. Results

In this section, we present the results of the proposed method. First, we describe the procedure in which the data were collected. Then, we describe the results regarding the auto-labeling method, and finally, we describe the results of the gait phases classification.

### 3.1. Measured Data

All three participants completed the study protocol successfully, generating seconds more than 15 min of measurements collected at 100 Hz, equivalent to more than 90,000 samples per participant and 270,000 in total. As mentioned in the Section 2.5.2, all instances labeled as “NC” were eliminated from the total data, and also any sample that had a failure during transmission. This information is summarized in Table 2, where it is shown that after preprocessing the samples, the dataset available for training and testing the classification models was reduced on average to 50.81%, compared to the original data, in which more than 99.64% of the deleted data corresponded to “NC” instances.

Figure 6 shows a sample of the textile pressure sensor signals located on the knees while walking three steps.

### 3.2. Labeling Method

The automatic label generation method explained in Section 2.5.1 was applied for the three participants while recording the sensor signals in text files and their movements in video recordings simultaneously. From these videos, it was possible to correct misassigned classes manually. The difference between the automatic labeling system’s predictions and those observed in the recordings was used to calculate the system’s precision for each class. In other words, the number of true positives was divided by the sum of false positives and true positives, resulting in the results shown in Table 3, where percentages for each class and participant are shown. In addition, precision was calculated for each participant based on the correctly predicted classes over the total. The results show 87.66% for participant 1, 90.9% for participant 2, and 92.81% for participant 3.

### 3.3. Classification of Gait Phases

The data from the textile pressure sensors and those from the IMU were separated into two attribute-relation file format (ARFF) files for each participant, generating a total of eight databases. Each database was used to train and evaluate three classification models, whose results are presented next.

#### 3.3.1. Precision

Precision was the key metric observed out of the 24 evaluations (three classification models for eight databases). It measured the percentages of correctly predicted cases over the addition of false positives and true positives cases. Algorithms were tested over the datasets generated from each participant, individually, and also over a single dataset, which included all participants’ signals. Table 4 shows the precision of the trained classification models, in which the test set corresponded to 20% of the dataset’s total, and 80% corresponded to training. The comparison of the precision achieved by the models trained with the textile pressure and the IMU sensors is as follows. In the case of the individual datasets (participants 1–3), it can be noted that the precision of random forest (RF) combined with the textile sensors was 2.28 ± 1.23% less than that of the IMU. For time series forest (TSF), on the other hand, it was 1.97 ± 0.62% less than that of IMU. Conversely, the model trained with the Mr-SEQL algorithm showed a precision for the IMU 50% below that of the textile sensors. Nevertheless, the Mr-SEQL algorithm generally presented lower precision values for the textile pressure sensors than the other classification models, achieving an average precision of 81.44%. In contrast, for RF and TSF, the average values were 90.82% and 90.43%, respectively.

Regarding the results obtained over the databases that included mixed samples of all participants, it can be observed that despite their different physical characteristics, the models obtained prediction precisions very similar to those obtained individually aver each test participant. In that case, the worst result was still for the Mr-SEQL model, having a 78.97% precision. In contrast, for RF and TSF, the values were slightly better than those averaged from their individual cases, with 91.22% and 90.53%, respectively.

#### 3.3.2. Confusion Matrix

The confusion matrices were estimated for each of the 24 classification models trained and tested in this research. Figure 7 shows the confusion matrices obtained with the three classification models for the data from the textile pressure sensors measured with participant 1, where the rows correspond to the true labels, and the columns to the predictions made for each of the models. A consistent result was presented in the three confusion matrices regarding the class that generated most of the erroneous classifications. Here, “RSTLST” was similarly confused by the model with the classes “RSWLST” and “RSTLSW”, obtaining an average error among the three models for “RSWLST” of 8.79 ± 2.49, and 9.03 ± 1.61 for “RSTLSW”. On the other hand, the two phases in which at least one leg was swinging were distinguished very well in all classification models, with an average error of 0.5 ± 0.48.

## 4. Discussion

This research aimed at detecting the gait phases of test participants by combining homemade textile pressure sensors sewn to elastic pants with classification models. This approach would permit the continued use of more comfortable gait analysis tools at home, especially for elderly people. The results are discussed as follows.

### 4.1. Results

The classification results demonstrated the feasibility of using such sensors, which—despite being low-cost, easy-to-manufacture and, therefore, variable in terms of sensitivity—achieved a precision of around 90–92% when combined with random forest or time-series forest classification algorithms. This result is slightly less than the 92–93% obtained from an IMU sensor using the same algorithms, even though such sensors are significantly more sophisticated devices, e.g., an IMU comprises precise internal MEMS instruments, such as a three-axial accelerometer, a three-axial gyroscope, a three-axial magnetometer, and a barometer, which can measure height changes of the test participants. We chose such a device comparison since it is the gold standard in gait analysis with wearable devices. We also evaluated a Mr-SEQL algorithm, which did not obtain satisfactory results for either the textile sensors (<84%) or the IMU sensors (<38%).

The evaluation also shows that, despite the physical differences among the three test participants, precision for the classification models on the textile sensor data remained similar for all participants, combined or individually. This result indicates that using the proposed workflow, the solution can be scaled to more participants, and an individual calibration process would not be required. Furthermore, one of the most significant benefits of using machine learning techniques on gait analysis applications is that precise and consistent sensors are unnecessary. For example, as shown at the beginning of the results section, the electrical resistance of the textile sensors for the same activity, and having the same test participant, might vary significantly from one step to another. However, it was possible to adequately evaluate their behaviors and detect the participants’ gait phases. In contrast, relying on individual parameters, such as signal thresholds, requires individual calibration processes for each sensor whenever a test is conducted since the pressure conditions change with the person’s position and physical characteristics.

Our results show that a single IMU sensor performs similarly to ten textile sensors. It can be argued that the former is less expensive and more comfortable than the textile alternative. Nevertheless, some advantages of textile sensors are that they can be better integrated into clothing when used at multiple points than multiple IMU sensors, which require rigid printed circuits. Additionally, they are easy and fast to manufacture. For some applications where the subject’s presence needs to be detected, textile sensors are preferable to IMUs, since they measure pressure instead of acceleration.

In this study, we used an arbitrarily large number of sensors to generate a sufficiently dense sensing spatial coverage on the pants. However, further analysis of the contribution of each body part should be performed to reduce the number of sensors and define their optimal location for future applications. Moreover, textile sensors have some aspects to take care of. For example, to properly distinguish the pressures produced by body movements, textile sensors must be integrated into stretch clothing that adjusts well to the body, which might be inconvenient in hot latitudes. Moreover, designers must match the conductive threads’ elasticity with that of the expandable fabric pants, as significant differences will tend to produce thread breaks whenever the participants bend their legs beyond a certain angle.

On the other hand, the data labeling system used in this research takes advantage of the BlazePose keypoint detection and pose estimation model fed from webcam images. The model tracks the toe and heel key points to estimate the feet velocities, among others. These data are relevant for the gait phase detection, obtaining a precision of 90.46 ± 2.13% in the label classification, which is remarkable for a classification system that bases its operation only on thresholds. Furthermore, for all of the test participants, the class in which both legs were in the stance phase showed a reliable classification, achieving an average precision of 99.57 ± 0.17%. On the contrary, the performance was less effective for classes having one of the legs in the swing phase, achieving an average precision of 82.14 ± 5.41% for the “RSTLSW” class, and 88.41 ± 0.97% for the “RSWLST” class. The results show that the proposed method can be used for labeling in real-time in supervised machine learning applications where creating datasets of body sensors is required. For example, body MEMs sensors, body stretch/force/pressure sensors, external microwave sensors, LiDAR sensors, or any model that requires matching a given pose with a time-series signal for its training. Nevertheless, the method still has space for improvements, and visual inspections of the tests are necessary to correct classification errors. In the setup used in this studio, the errors mainly occurred whenever the leg facing the camera occluded the other leg. In addition, given that the participants required specific and regular walking speeds, errors occurred in the labeling process whenever participants mismatched that speed.

### 4.2. Study Limitations

The study protocol used in this research was limited to a laboratory having a ceramic floor and a regular walk, so the results did not include other types of contexts where the participants may have presented different behaviors. Moreover, the available space and the experimental setup in which the camera was fixed relative to the path that the participants followed contributed toward generating an important number of NC instances, which were not considered in the training and testing process of the ML models. In addition, all the participants walked at a self-selected speed, which was problematic since this variability directly impacted the reliable measurements, contributing to the error in the whole phase detection process. This sensor response differences could be easily evaluated using the same workflow proposed in this article but performing the walk trials using a treadmill.

Finally, our study’s participants were approximately 27-year-old healthy men. A more varied population should be considered, with a broader range of physical characteristics to train and test the classification models, to obtain a more concrete answer to the question on whether a calibration process is required or not.

### 4.3. Future Work

Although the predictions achieved with the proposed workflow show promising results, real gait analysis applications require estimating more spatiotemporal parameters, e.g., the different sub-phases of the swing and stance phases. In future work, we will extend the number of phases. Moreover, we will exploit the potential of the pants to estimate the flexion of the knees, being able to extract more helpful information for the diagnosis and evaluation of gait disorders. Moreover, as the experiments were performed under limited conditions, more measurements will be conducted, evaluating different environment conditions and optimizing the helpful information extracted from them. Finally, another challenge in future work will be to implement this type of gait phase detection algorithm locally and in real-time on an embedded device, thus avoiding connection with cloud services.

## Figures and Tables

**Figure 1 sensors-22-02825-f001:**
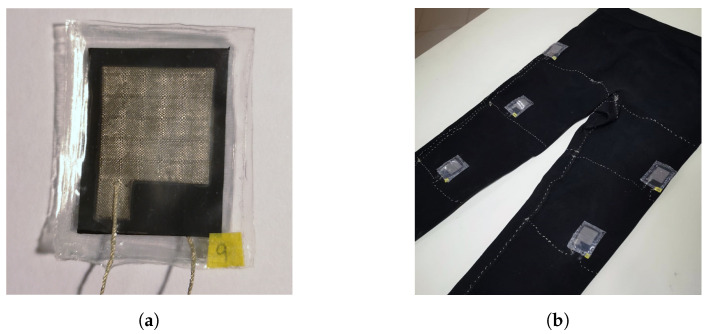
(**a**) Sample of the textile pressure sensors manufactured in the laboratory; (**b**) Textile pressure sensors attached to the expandable fabric pants.

**Figure 2 sensors-22-02825-f002:**
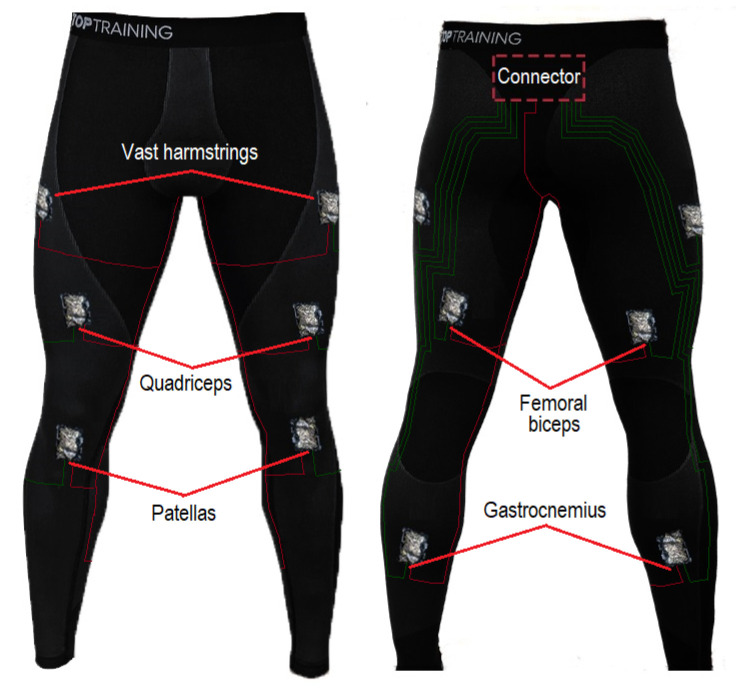
Location of the textile pressure sensors on the expandable fabric pants.

**Figure 3 sensors-22-02825-f003:**
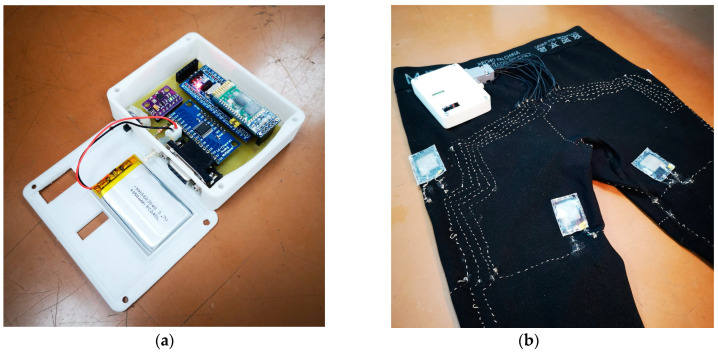
(**a**) Acquisition card in a 3d-printed case; (**b**) Location of the acquisition board on the expandable fabrics pants.

**Figure 4 sensors-22-02825-f004:**
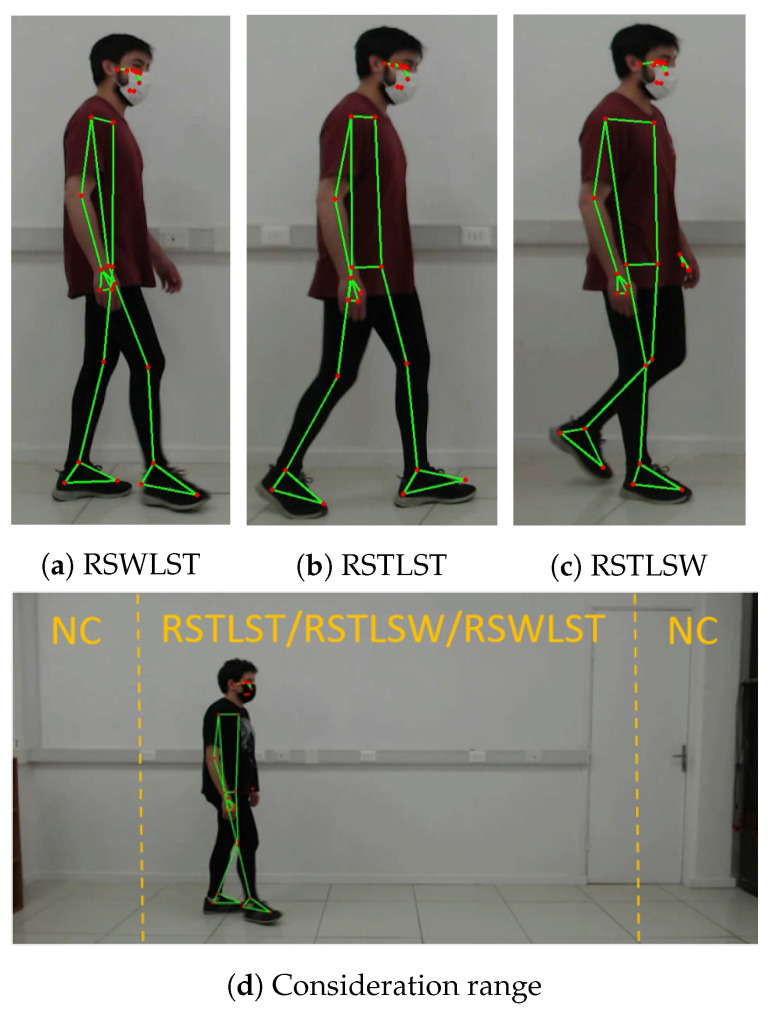
(**a**) Example of the right leg swing and left leg stance class; (**b**) Example of both leg stance classes; (**c**) Example of the right leg stance and left leg swing class; (**d**) Data consideration range in the camera frame of the lab setting.

**Figure 5 sensors-22-02825-f005:**
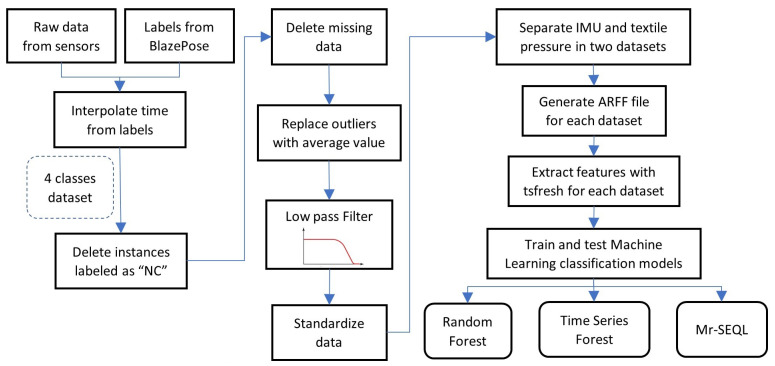
Workflow used to test and train the classification models from the raw data and the generated labels.

**Figure 6 sensors-22-02825-f006:**
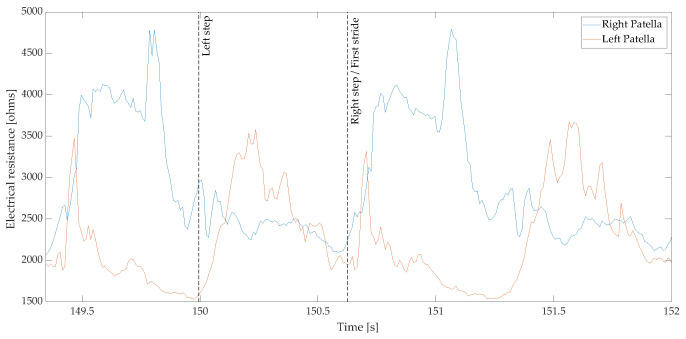
Sample of collected textile pressure sensor raw data located over a participant’s patellas during two strides.

**Figure 7 sensors-22-02825-f007:**
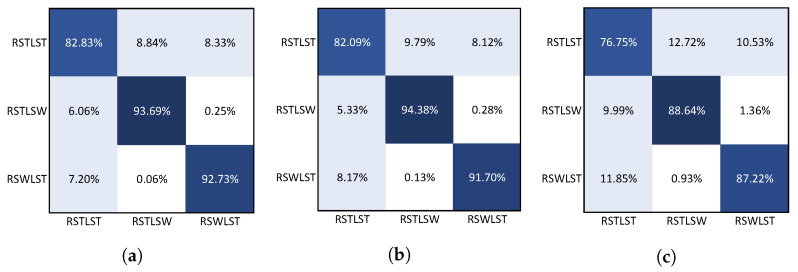
Confusion matrix of models trained with textile pressure sensors data from participant 1. (**a**) Random forest; (**b**) Time-series forest (TSF); (**c**) Mr-SEQL. Dark colors indicate proximity to 100%.

**Table 1 sensors-22-02825-t001:** Physical characteristics of the participants in the study.

	Participant 1	Participant 2	Participant 3	All Participants
Age	27	29	24	27
Height	1.61 m	1.76 m	1.74 m	1.70 m
Weight	66 kg	75 kg	83 kg	75 kg
BMI	25.3	24.3	27.4	25.7

**Table 2 sensors-22-02825-t002:** Number of samples obtained in each stage of the process and percentage used.

	Participant 1	Participant 2	Participant 3
Raw data	90,455	92,201	94,092
Data without NC	46,357	47,170	47,272
Pre-processed data	46,192	47,151	47,244
Percentage used	51.06%	51.16%	50.21%

**Table 3 sensors-22-02825-t003:** Precision percentage of the labeling system for each class and participant.

Class	Participant 1	Participant 2	Participant 3
RSTLST	99.53	99.79	99.39
RSTLSW	76.81	80.06	89.55
RSWLST	87.53	89.76	87.94

**Table 4 sensors-22-02825-t004:** Precision percentage achieved with each participant and type of sensors.

	Participant 1		Participant 2		Participant 3		All Participants	
Algorithm	IMU	Textile	IMU	Textile	IMU	Textiles	IMU	Textile
Rand Forest	92.77	89.91	93.29	92.72	93.24	89.84	92.00	91.22
Time Series F.	92.05	89.57	93.35	92.25	92.80	89.47	92.85	90.53
Mr-SEQL	36.62	84.34	38.05	81.87	40.70	78.11	36.80	78.97

## Data Availability

The data used for this investigation are available upon request from the corresponding author. The data are not public due to privacy and ethical restrictions. Source code available here [36].

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
