# Peer review of "Detection of Human Gait Phases Using Textile Pressure Sensors: A Low Cost and Pervasive Approach"

_sensors, 2022, doi:10.3390/s22082825_

Round 1
Reviewer 1 Report
The paper is very interesting and correct developed. The paper evaluates the use of low-cost homemade textile pressure sensors to recognize gait phases. The authors results show that textile sensors successfully detect the gait phases with a hight precision. The authors use professional MATLAB software in yours work and machine learning algorithms. The paper is characterized by a well-made literature review and a precisely described course of the experiment. Although the publication is more engineering than scientific, I highly recommend the work for publication in Sensors. In the publication, the presented solution may bring many utilitarian results. Also I would like to recommend some publications in this field to authors (see below). The recommended works also concern the research and use of sensors based on textile materials and have a number of different applications. The overall results show the feasibility of using textile pressure sensors for human gait recognition. I recommend authors to extract unambiguous and scientific conclusions resulting from the proposed technique and promotion this method of detect of Human Gait Phases Using Textile Pressure Sensors.
https://doi.org/10.1051/matecconf/201925209005 (https://www.matec-conferences.org/articles/matecconf/pdf/2019/01/matecconf_cmes2018_09005.pdf)
DOI: 10.15199/48.2017.12.28 (https://www.researchgate.net/publication/321793427_Odpornosc_warstw_metalicznych_stosowanych_w_systemach_tekstronicznych_na_deformacje_mechaniczne)
Reviewer 2 Report
It is an interesting study and may have many potential and practical applications. I have found no substantive deficiencies in the manuscript. Naturally, some improvements should be made before it can be accepted for publication:
1) The paragraph from line 116 to 120 can be deleted. It provide no substantive content.
2) The paragraph from line 122 to 126 can also be deleted.
3) Square brackets are not required for units. "17 [s]" should be "17 s", "10 [kPa]" should be "10 kPa", etc.
Reviewer 3 Report
This paper proposes an application of the textile pressure sensor that the authors have previously proposed for gait analysis.
However, from the contents shown in the experiment, both novelty and usefulness as an academic paper seem to be low. It would be better to reconsider the contents of the proposal and the experiment.
- The results didn’t give sufficient explanation as to why ten textile pressure sensors were used.
The experiment in this paper only classifies the gait phases (RSTLST, RSTLSW or RSWLST classifications).
Wouldn't this rather give better results with two sensors in the insoles of both feet?
Shouldn't the experiment be conducted to estimate the flexion angle or foot lift height?
- Please clarify the advantages over other studies.
There are flexible motion capture systems, e.g., StretchSense's products, or research on whole body motion estimation using soft sensors and machine learning [D.Kim, et al. 2018].
If low-cost is an advantage, then the following is of concern.
-
- Why is costing important?
- How cost-effective is it compared to other studies?
Wouldn't accuracy be a higher priority than cost in medical applications?
* D. Kim, et al. Deep Full-Body Motion Network (DFM-Net) for a Soft Wearable Motion Sensing Suit. IEEE/ASME Trans. on Mechatronics, 2018, 24.1: 56-66. 3.
- Is the scope of the experiment too limited?
It is understandable that half of the dataset was excluded as NC, but the fact that the analysis can only be performed under limited conditions makes it less useful.
Real-time labeling is mentioned as the second contribution, but from the following, I cannot find any application other than this study.
-
- The limited situation of capturing a stable gait of about 0.7 [m/s] from the side.
- Only three classifications, RSTLST/RSTLSW/RSWLST, are available.
Even for the use in this study, for example, if a motion capture facility that is not a simple camera is used, and the method of labeling RSTLST/RSTLSW/RSWLST in a circular walking situation, it would be almost NC-free.
The detailed description of the procedure, the release of the source code on GitHub repository, and the provision of the dataset are appreciable.
Round 2
Reviewer 3 Report
I judged that the experiment content did not conform to the publication with only the three classifications of RSWLST, RSTLSW, and RSTLST.
The situation with COVID-19 is unfortunate, but my guess is that it is possible to find the knee joint flexion angle from the existing BlazePose data.
Even if the result is low accuracy, can you show the possibility that the proposed suit can detect other parameters such as flexion angle?
The Abstract also highlights the description of the accuracy comparison between IMU and gait phase.
However, it is a simple task of 3-class classification, and a single IMU on the waist is disadvantageous to compare to 10 pressure sensors.
A single IMU with comparable results would have the following advantages over the suit.
- Hygienic
- Low-cost
- Patient’s freedom of movement when walking
There is also an IMU-based motion capture suit called e-skin MEVA, which has sufficiently improved the problem of the stiffness and size of the holding printed circuit board, and is washable.
The proposed pressure sensor suit is still in the experimental stage, but ideally, it would be very valuable if it could achieve equivalent performance compared to such products at a much lower cost.
Round 3
Reviewer 3 Report
The essential part of the proposal, that there is a mismatch between the proposed suit and the experiment, has not been corrected.
If the principle is to "keep it simple," why is it necessary to put sensors in ten locations?
However, it is understandable that the suit was created with future potential in mind.
Before using ten sensors, it may be possible to classify three classes using only the sensors on both patellas, and if it is difficult, I am curious about the consideration of why, but I look forward to future research.
(Around 414th line)
Also, by using pressure sensors instead of an IMUs, it perhaps possible to find other applications other than motion capture, such as pressure ulcer prevention or prevention of high-load postures.
Author Response
Dear reviewer,
Please find below the reply to the comments. We appreciate the interest and valuable feedback, which positively impacted the manuscript's quality. We modified the manuscript's introduction in line 79 to include the reviewer's last comments.
Question: The essential part of the proposal, that there is a mismatch between the proposed suit and the experiment, has not been corrected.
If the principle is to "keep it simple," why is it necessary to put sensors in ten locations?
However, it is understandable that the suit was created with future potential in mind.
Before using ten sensors, it may be possible to classify three classes using only the sensors on both patellas, and if it is difficult, I am curious about the consideration of why, but I look forward to future research.
(Around 414th line)
Answer: Thanks again for the valuable comments. We believe that, in our case, the number of sensors does not imply more complexity, so there is no mismatch. As stated in line 146 of the paper, the ten sensors were manufactured using only office supplies and low-cost materials, so the method can still be classified as simple. As pointed out, the suit was created with ten sensors as an arbitrarily large number with future potential in mind and to reduce that number in future work, as mentioned in lines 412-414, after individual sensor analysis.
Question: Also, by using pressure sensors instead of an IMUs, it perhaps possible to find other applications other than motion capture, such as pressure ulcer prevention or prevention of high-load postures.
Answer: We appreciate the suggested applications. Measuring the exact spot where excessive or permanent force is being applied to prevent ulcers is an excellent use case for force textile sensors. In that case, textile sensors might be attached to the patient’s underwear or mattress; however, when attached to the underwear, force sensors can be more confidently correlated with a particular affected body part. IMUs, as suggested, are not appropriate for this application, as they measure acceleration, not pressure. Other similar uses of textile sensors might include lifestyle applications to let users know if they have been seated too long (during office work or driving).
Action: We include this comment in the introduction (line 79).